# Nutrients cause grassland biomass to outpace herbivory

E. T. Borer ⓘ et al.[#]

Human activities are transforming grassland biomass via changing climate, elemental nutrients, and herbivory. Theory predicts that food-limited herbivores will consume any additional biomass stimulated by nutrient inputs ('consumer-controlled'). Alternatively, nutrient supply is predicted to increase biomass where herbivores alter community composition or are limited by factors other than food ('resource-controlled'). Using an experiment replicated in 58 grasslands spanning six continents, we show that nutrient addition and vertebrate herbivore exclusion each caused sustained increases in aboveground live biomass over a decade, but consumer control was weak. However, at sites with high vertebrate grazing intensity or domestic livestock, herbivores consumed the additional fertilization-induced biomass, supporting the consumer-controlled prediction. Herbivores most effectively reduced the additional live biomass at sites with low precipitation or high ambient soil nitrogen. Overall, these experimental results suggest that grassland biomass will outstrip wild herbivore control as human activities increase elemental nutrient supply, with widespread consequences for grazing and fire risk.

---

[#]A list of authors and their affiliations appears at the end of the paper.

Grasslands are a critical sink for atmospheric carbon, a key energy source for terrestrial food webs, and a vital resource for human food and fuel production[1–3]. Grassland plant biomass is controlled by interdependent factors that vary in space and time[4], including climate[3,5], the availability of growth-limiting resources, such as nitrogen and phosphorus[6,7], and plant interactions with herbivores[8]. However, human activities are altering these processes[9–11]. For example, regional signatures of temperature and precipitation are changing[12], and anthropogenic nitrogen deposition has increased dramatically since the start of the 20th century[6,13,14]. Growing evidence suggests that widespread, but regionally variable, eutrophication of terrestrial ecosystems[6], and alteration of climate are changing global grassland productivity[15–17]. Yet, estimates of nutrient limitation of biomass production are commonly performed in ecosystems without accounting for the effects of herbivores[18,19]. This hinders our ability to evaluate the generality and magnitude of herbivore control of plant biomass. Notably, while grassland biomass production is critically important for services, including animal forage, soil health, and atmospheric carbon capture[3], reduced consumer control of biomass in a eutrophic world could, for example, reduce plant biodiversity[20] or increase fuel load and fire severity[21].

Simple equilibrium theory suggests that herbivores should be able to consume the additional plant production stimulated by elevated nutrient supply[22–24]. In particular, "consumer-controlled" theory predicts that when consumers are limited by their food resources, consumption will increase to counter any additional production, leading to no net change in plant biomass[22] (Fig. 1a). Although a wealth of experiments in marine and freshwater ecosystems demonstrate that herbivory can counterbalance increased primary production due to eutrophication[25–27], few studies have simultaneously manipulated terrestrial soil resources and herbivory by large vertebrates to test these predictions[28]. In a comprehensive meta-analysis summarizing nearly 200 experiments that concurrently manipulated both nutrient supply and herbivores, only 4% (eight studies) were in herbaceous-dominated terrestrial ecosystems and, of these, only four studies examined the effects of vertebrate herbivores[25]. These few grassland studies, generally lasting <3 years, suggest that herbivores have little impact on biomass, and fertilization can increase biomass even in the presence of herbivores[18,25]. Although data poor, this trend is more consistent with alternative "resource-controlled" theory that predicts increasing plant biomass along a productivity gradient even in the presence of herbivores[29–31]. Importantly, this theory predicts that herbivores will consume a constant proportion of plant biomass, regardless of environmental productivity (Fig. 1d).

A large body of theory has examined the consequences of a variety of realistic mechanisms that alter herbivore–plant interactions under eutrophic conditions[25,31], potentially shifting plants from resource control to consumer control[25,31,32]. For example, fast-growing, highly nutritious plant species with low investment in defense often dominate in high resource environments[33], which could lead to greater herbivore control of biomass in high resource environments than in conditions of low nutrient supply, as predicted by consumer-controlled theory (Fig. 1a)[25]. In contrast, theory that considers herbivore dietary specialization or selectivity suggests that herbivores will reduce the abundances of preferred plant species more in high resource environments. Compositional turnover due to this selective feeding can lead to reduced herbivore control of biomass with increasing productivity, ultimately resulting in proportional consumption (Fig. 1d) or even reduced consumption (Fig. 1g)[25,31,32,34–36]. In addition, the high among-study variability observed in plant biomass control by vertebrate herbivores[25,34] is likely governed by context-dependent responses, varying with precipitation[31,34], or other regional climatic, edaphic, or biotic gradients that could alter plant biomass control, but are not effectively characterized in a meta-analysis. Thus, there is a clear gap in existing data, but theory provides a guide for testing whether—and under what conditions—herbivores will control the accumulation of biomass in modern, eutrophic grasslands, where most wild grazer communities have been extensively modified and often driven to reduced population sizes by humans[11].

We test the "consumer-control" hypothesis that vertebrate herbivores in grasslands consume the additional primary production due to eutrophication[24,25], using an experiment replicated at 58 grassland sites spanning six continents. Our factorial experiment manipulates elemental nutrients (nitrogen, phosphorus, potassium, and micronutrients) and vertebrate herbivores larger than ~50 g (see "Methods" section and ref. [37]), allowing us to test for the hypothesized interaction between nutrients and herbivores in controlling grassland biomass[25]. Each year in each plot, we used standard methods[37] to measure aboveground live biomass. We test for non-additivity in log space, or a non-proportional dependence of herbivory on nutrient addition. Thus, a positive interaction indicates that herbivores consume proportionally more biomass under fertilized conditions compared to control, the hypothesis arising from the simplest consumer-resource models[22,24] (Fig. 1a–c). No interaction in log space indicates herbivores track increased biomass production with a proportional increase in consumption[29–31] (Fig. 1d–f), and a negative interaction indicates herbivores consume proportionally less under fertilized conditions[25] (Fig. 1g–i). We test whether a variety of biotic and abiotic factors are associated with the strength of the interaction between nutrients and herbivores. Importantly, this includes determining the role of herbivore impact by additionally testing whether plant biomass control under eutrophication increases with increasing wild herbivore abundance and diversity or is greater at sites with domestically managed livestock than those with only wild herbivores.

This distributed experimental work demonstrates that, on average, herbivores in grasslands around the world remove a constant proportion of fertilized biomass. Thus, grassland biomass accumulates with fertilization, even with wild herbivores present. However, these results are context dependent. In particular, in locations where wild herbivores remain abundant or domestic livestock are also present, herbivores tend to keep up; they remove proportionally more biomass under fertilized conditions than under ambient conditions. Herbivore effects also vary along biogeographic gradients. For example, with increasing ambient soil nitrogen, plant biomass shifts from resource- to consumer-controlled, and herbivores keep up with fertilized biomass production in sites with low precipitation.

## Results

**Testing for an interaction between fertilization and fencing.** This distributed experiment, performed at 58 grasslands sites on six continents, and including sites with wild vertebrate herbivores where fencing more than doubled aboveground live biomass (Supplementary Fig. 2), provided a strong test of the ability of vertebrate herbivores to consume the additional plant biomass produced in response to environmental eutrophication. Nonetheless, the experiment provided no evidence for an overall interaction between fertilization and fencing for most of the study duration ($P > 0.05$), indicating constant proportional biomass removal under both ambient and elevated nutrient conditions (Figs. 1d–f and 2).

The fertilizer-induced increase in biomass inside fences was marginally smaller than the increase induced in the presence of

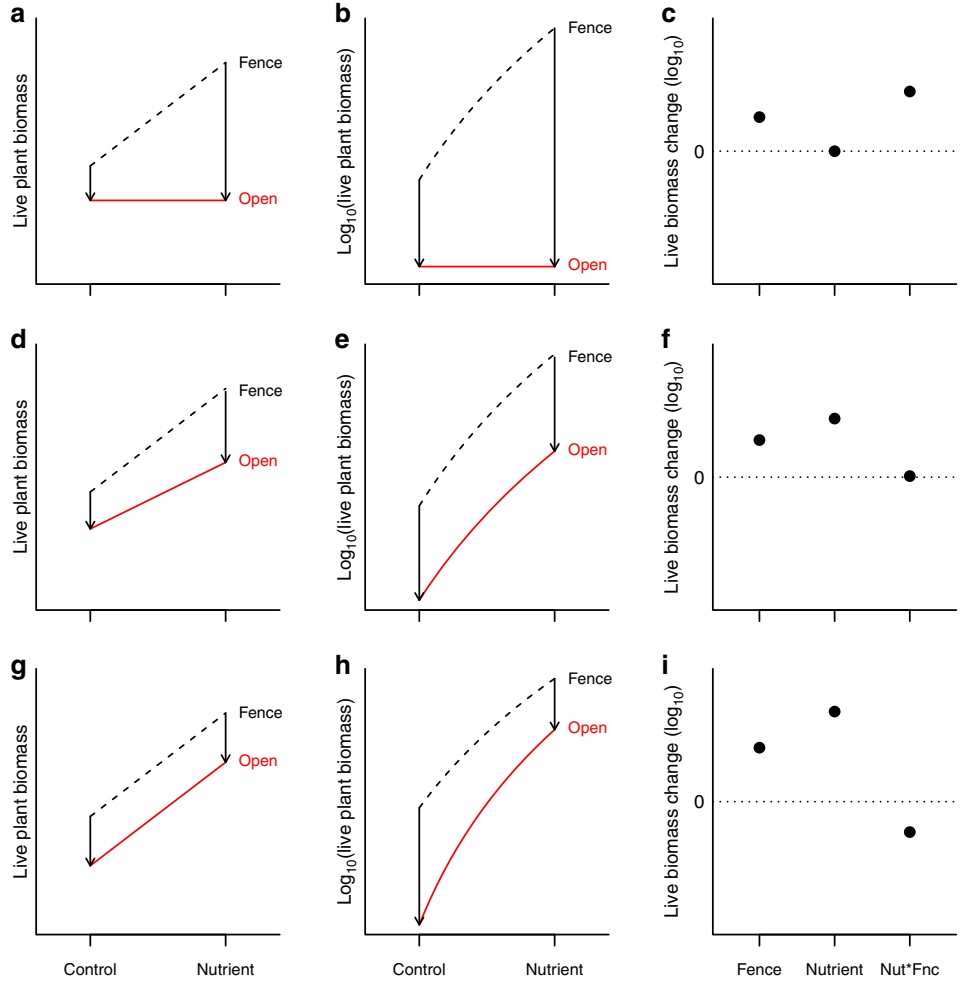

**Fig. 1 Predictions for plant biomass with increasing environmental productivity.** Arrows indicate the predicted difference in biomass at ambient (Control) and elevated (Nutrient addition) productivity in the presence (Open) and absence (Fence) of herbivores. Predictions for plant mass with increasing productivity when herbivores **a** consume all additional biomass produced (see ref. [24]); **d** consume a constant proportion of biomass (see ref. [29]); **g** consume a constant amount of biomass. Panels **b**, **e**, **h** show these predictions for log(biomass). Panels **c**, **f**, **i** show the difference (arrow lengths (fence and nutrient) and difference of arrow lengths (Nut*Fnc) in **b**, **e**, **h**), and visualize these hypotheses as they would look in the factorial experimental test.

large vertebrate herbivores in the longest-running sites (6.6% reduction at 8–10 years, Fig. 2 and Supplementary Table 3c), pointing to a slightly larger proportional impact of herbivores under ambient than fertilized conditions (Fig. 1g). However, across all sites, herbivore impacts on plant biomass with elevated nutrient supply were predicted by their impacts under unfertilized, ambient conditions (slope = 0.85; $r^2$ = 0.20, Supplementary Fig. 2).

**Mean effects of fencing and fertilization.** In spite of the absence of support for an interaction between fencing and fertilization in controlling grassland biomass, both treatments altered grassland biomass. Across sites spanning six continents, both exclusion of vertebrate herbivores ($P < 0.001$, Supplementary Table 3a) and fertilization ($P < 0.001$, Supplementary Table 3a) increased aboveground biomass, with vertebrate herbivore exclusion leading to a 12% average increase in biomass by year 2, and fertilization leading to an average 58% biomass increase (Fig. 2 and Supplementary Fig. 1).

**Temporal trends in fertilization and fencing.** Because our experiment was replicated for 2–10 years at all sites, we tested the hypothesis that the variation in herbivore effects observed in past studies can be explained by study duration[25], by comparing our

full range of sites to the subset of 42 sites with 5 or more years and the subset of 24 sites with 8 or more years of continuous experimental manipulations. The subset of 24 sites with 8 or more years of data demonstrated that nutrients ($P < 0.001$) and herbivore exclusion ($P < 0.001$) led to a persistent increase in biomass of similar magnitude to the shorter-term effect across all sites (Fig. 2 and Supplementary Table 3b).

**Testing for an interaction contingent on herbivore type, herbivore biomass, or herbivory intensity.** Vertebrate herbivores consumed much more of the fertilization-induced biomass at sites where domestic and wild herbivores were both present. In particular, for the subset of eight sites with a mix of domestic livestock and wild herbivores (Supplementary Table 1), the biomass increase due to fertilization inside fences was 41% greater than expected from the independent effects of these treatments ($P = 0.006$, Supplementary Table 3d and Supplementary Fig. 3).

We additionally tested whether herbivore control of fertilized biomass increased either with an herbivore index based on site-level expert knowledge[38] that quantified herbivore impact intensity and frequency or with modeled herbivore biomass[23,32,39]. Across all sites, the additional fertilized plant biomass was removed where the herbivore index was very high ($P = 0.01$, Fig. 3a and Supplementary Table 4a). However, at most sites, represented by low to

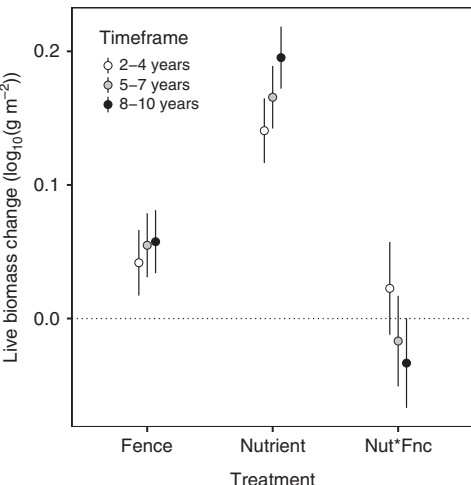

**Fig. 2 Treatment effects on aboveground grassland biomass were similar for sites with 2–4 years of response data (n = 58, open circles), at least 5 years (n = 42, gray), and at least 8 years (n = 24, black).** Error bars represent 95% confidence intervals. Main effect values represent model mean differences of the experimental treatment from the control after controlling for site and year as random effects. The nutrient by fence interaction ("Nut*Fnc") is compared to the sum of the effect of nutrients alone and fencing alone, such that zero indicates additivity.

intermediate herbivore index values, herbivores did not keep up with the additional fertilization-induced biomass. Modeled herbivore biomass did not predict the site-level biomass response ($P = 0.18$, Supplementary Table 4A). Because of covariation between the intensity and frequency of herbivores (herbivore index) and other site-level characteristics[34], the impact of herbivores on fertilized biomass was less apparent in models just including site-level climate data (Supplementary Table 4b), but was clear when ambient site-level nutrients also were included (Supplementary Table 4c). Although the herbivore index results were consistent with the analysis of sites with domestic livestock (Supplementary Table 3d and Supplementary Fig. 3), the sites with the largest herbivore index values were distinct from those with domestic grazers; only two of the top ten sites with the greatest index values had domestic grazers (Supplementary Table 1).

**Testing for an interaction contingent on abiotic and biotic characteristics.** The sites in this experiment spanned a wide range of ambient edaphic (e.g., 90–17,160 p.p.m. soil N), climatic (e.g., 246–1,877 mm mean annual precipitation), and biotic (e.g., 1–31 plant species per m$^2$) characteristics, providing insights into the contingency of fence effects on local abiotic and biotic factors. This experiment demonstrated that the impact of fencing varied in strength with edaphic characteristics and climate. In particular, nutrients increased biomass in the presence and absence of herbivores, whereas excluding herbivores had little effect at sites with low soil N, but had an increasingly positive effect on plant biomass with increasing soil N ($P = 0.006$, Fig. 3b and Supplementary Table 4c). Herbivores also had the greatest effects in fertilized plots at low precipitation sites (Fig. 3c and Supplementary Table 4b). In contrast, the magnitude of herbivore effects on biomass was not associated with either plant species richness or species temporal turnover ($P \gg 0.05$).

## Discussion

In contrast to earlier syntheses of existing data[25,28], the results of this multisite replicated experiment demonstrate that grassland biomass around the world is limited by vertebrate herbivores, as well as nutrients, revealing striking similarity to the main effects of consumers and resources in more extensively studied marine and freshwater ecosystems[18,25–27,40]. However, in spite of a very wide range of biomass responses among sites—including sites with wild herbivores where fencing more than doubled biomass—vertebrate herbivores did not, in general, keep up with fertilized biomass production. Thus, the results of this study provide little support for the central hypothesis that wild terrestrial herbivores remove the additional biomass produced in fertilized plots, for example, due to greater nutritional quality or palatability[41]. Instead, evidence from this experiment points to constant proportional biomass reduction by herbivores, indicating that herbivores consume more plant biomass under fertilized conditions; nonetheless, fertilized grassland biomass accumulation outpaces herbivory.

Although wild herbivores did not, on average, consume the additional biomass produced with fertilization, the interaction between nutrients and herbivores increased with site-level intensity of herbivory[23,24,34], highlighting the context-dependence of resource and consumer control of plant biomass[22,29–31]. In particular, these globally distributed experimental results suggest that herbivores will, on average, consume a constant proportion of biomass (Fig. 1d–f). In contrast, in environments where wild herbivores remain abundant[11] and in human-managed agricultural settings, herbivores can remove the additional biomass produced in future, eutrophic grasslands, consistent with simple consumer-controlled theory[22,24] (Fig. 1a–c). However, given that wild herbivore population sizes in many regions of the world are limited by factors such as habitat loss, hunting, and disease[11] rather than food supply, the increased plant biomass from eutrophication in grasslands without domestic grazing is likely to remain mostly unconsumed. Further, the observation that long-term nutrient deposition has led to increased live plant biomass in the presence of herbivores across a wide range of global grasslands[15] is consistent with the prediction that arises from this experiment: in most grasslands, herbivores are not keeping up with increased biomass production. While unavailable for the current study, site- and plot-scale measurements of the rates of plant productivity and consumption rates by vertebrate herbivores would provide additional insights into the global variation—and likely future trends—in herbivore control of grassland biomass.

Although studies in some terrestrial ecosystems have found very long timescales necessary to detect plant biomass responses to herbivore exclusion[42], we found that 2–4 years of treatments led to similar conclusions about the overall effects of herbivores on grassland biomass, as did study durations of up to a decade. Thus, our extensively replicated experimental results additionally demonstrate that herbivore exclusion leads to a rapid increase and persistently elevated biomass through time, in contrast to previous data syntheses that found no overall effect of herbivores across studies and a general decline in the effects of herbivores on grassland biomass with time[25].

This multisite study, spanning a globally relevant range of edaphic and climatic characteristics, supports hypotheses that abiotic factors also determine the conditions under which herbivores consume the additional biomass from eutrophication[4,15,19,38]. Along biogeographic gradients of soil nitrogen, our results demonstrated that plant biomass shifted from resource- to consumer-controlled. These results are consistent with the theoretical prediction that biomass control will change along a gradient of environmental productivity[24,31] and that the plant species that dominate in high resource environments[33,38] are particularly susceptible to herbivore control of biomass[24]. This result from our multisite experiment clarifies that the

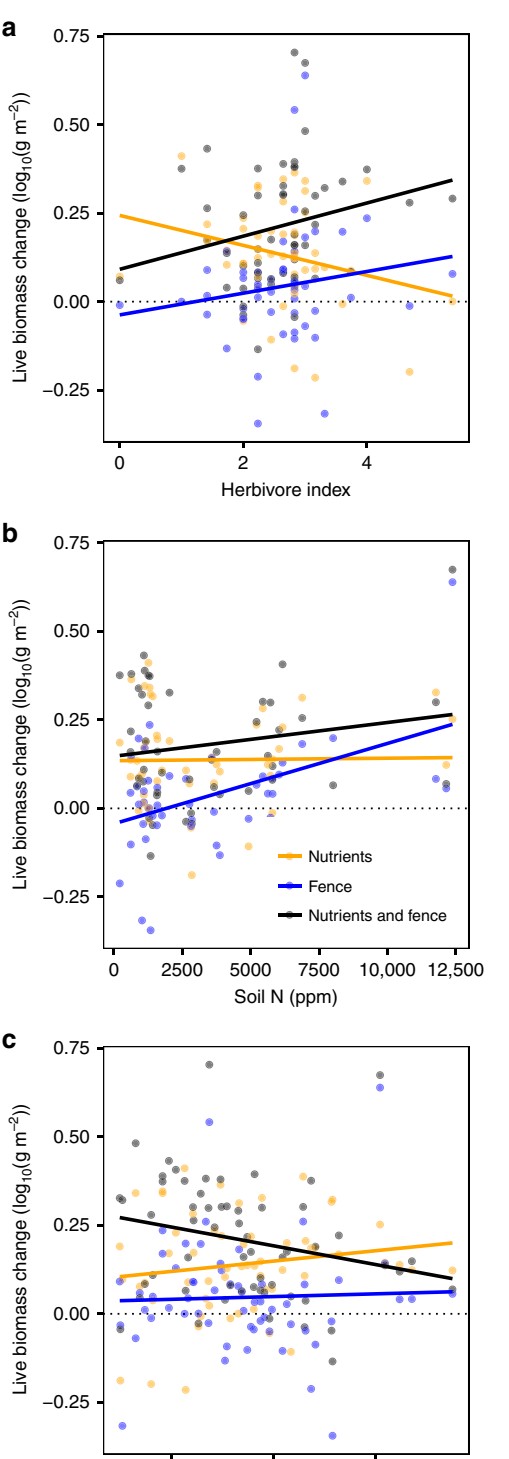

**Fig. 3 Herbivore control of grassland biomass varies with the biotic and abiotic environment.** Plots show the average site-level difference of each treatment from the unfertilized, unfenced control across a gradient of **a** herbivore impact intensity and frequency, **b** ambient soil nitrogen, and **c** mean annual precipitation. Gradients in panels **a**–**c** were identified from statistical models presented in Supplementary Tables 4a–c.

impact of herbivores on plant biomass is contingent on the abiotic characteristics of a site, providing explanation for the high variability in herbivore control of biomass that has been observed in meta-analyses[25,34].

The impact of herbivores also varied with climate, with greatest effects of vertebrate herbivores in fertilized plots at low precipitation sites. Taken together with the consistent proportional impact of fencing on biomass across the precipitation gradient, these results do not support the hypothesis of an increasing rate of predator control of herbivores or plant regrowth along a precipitation-induced productivity gradient[31,34]. Rather, the marked impact of herbivores on fertilized, but not unfertilized, biomass at low precipitation sites is consistent with differential impacts of herbivores on plant biomass due to palatability. In particular, a general pattern that has been documented in this experiment[38] is that when plant growth is limited by water, nutrients accumulate in plant tissues, whereas with increasing water availability, the nutrient to carbon content of tissues decreases with plant growth, even when nutrients are added[43]. Thus, the nutrient content of fertilized biomass tends to be higher than control conditions in dry sites, but similar to control in mesic sites[38], making fertilized plots in dry regions markedly more attractive to herbivores[44]. This result likely reflects a pre-ference response of herbivores to fertilized plots in regions with low precipitation. However, with nutrient addition at larger spatial scales, only if herbivore population growth is limited by the nutritional quality of resources in dry sites could this effect amplify with time. While population dynamics of herbivores in some low precipitation systems can vary with the quality of their resources[45], elevated nitrogen inputs can lead to increased standing biomass even in dryland systems[46], suggesting that this preference response in low precipitation regions may be weaker at the landscape scale. However, while decreased fertilization effects on biomass production in arid systems are often attributed to nutrient immobilization or water resource limitation[47], these results suggest that herbivory also may play a role. Taken toge-ther, these directly comparable experimental results, spanning nearly an eightfold range of precipitation, point to variation in resource quality as a likely mechanism underpinning the strength of herbivore control of eutrophic grassland biomass[31,38].

Measures of plant composition allowed us to test whether herbivore control of biomass production declined with increas-ing plant species diversity or compositional turnover in response to the treatments[31]. Although herbivores can increase compo-sitional turnover toward dominance by grazing-resistant or tolerant plant species[48], leading to reduced impacts of herbivores on plant biomass[25,31], our data demonstrate that neither plant species richness nor species temporal turnover was associated with the magnitude of the fencing effects. Past studies have found conflicting effects of herbivores on live biomass, including strongly increasing[49,50] or decreasing[50–52] biomass, possibly due to variable responses by the species present at the site. Even within sites, plant species turnover in response to eutrophication often is not predictable by functional group, frequently responding in a species-specific manner to the combination of consumers and plants present at the site[42,53,54]. This extreme context-dependence likely explains why species turnover rates do not predict biomass responses to treatments across the wide range of grasslands included in this experiment[31].

This work fills a key knowledge gap: previous syntheses have bemoaned the scarcity of manipulative experimental studies testing hypotheses about the generality of effects of vertebrate herbivores and nutrient supply on terrestrial biomass production[25,28,42]. This experiment demonstrates that both nutrient supply and vertebrate herbivory control aboveground biomass in the world's grasslands, with treatment effects appearing rapidly, and persisting or increasing for up to a decade. It also underscores the importance of context-dependent impacts of herbivores along biotic, climatic, and edaphic gradients[31], with herbivores counteracting the effects of nutrient addition,

particularly at sites with low precipitation and those with high ambient soil nitrogen or high grazing intensity. Thus, while the grassland biomass response to fertilization shifted to control by consumers at very high site-level abundance and diversity of herbivores, this was only one among several biotic and abiotic factors associated with aboveground plant biomass control in grasslands around the world.

These results reconcile the high variability of grassland responses to consumer and resource perturbations that has been documented among sites and studies[25,28,49–52], and demonstrate that these forces most often operate independently to control grassland biomass. In an era in which the challenges of climate change and catastrophic wildfires, driven by high fuel loads, are omnipresent[3,20,21], understanding the controls on grassland biomass is crucial. This distributed experiment provides a powerful demonstration that the stimulation of plant biomass production associated with increasing nutrient supply generally exceeds herbivores' consumption capacity in non-agricultural grasslands, with implications for future grazing, biodiversity, and fire risk management strategies in the face of continued anthropogenic perturbation of global nutrient cycles.

## Methods
A full factorial combination of large herbivore exclusion via fencing ("control" or "fenced") and addition of nutrients ("control" or "all nutrients") was applied to 5 × 5 m plots at 58 sites spanning six continents, as part of the Nutrient Network experimental collaboration (www.nutnet.org; Supplementary Table 1)[37]. Most sites had three replicate blocks, and all sites had collected 1 year of pretreatment data and 2–10 consecutive years of posttreatment data (Supplementary Table 1). All sites were located in the herbaceous vegetation ("grassland") representative of the region.

**Treatments.** Experimental design and treatments are detailed in ref. [37]. Nutrient treatments ("NPK") received: 10 g N m$^{-2}$ yr$^{-1}$ as time-release urea [$(NH_2)_2CO$], 10 g P m$^{-2}$ yr$^{-1}$ as triple-super phosphate, [$Ca(H_2PO_4)_2$], 10 g K m$^{-2}$ yr$^{-1}$ as potassium sulfate [$K_2SO_4$] and 100 g m$^{-2}$ of a micronutrient mix of Fe (15%), S (14%), Mg (1.5%), Mn (2.5%), Cu (1%), Zn (1%), B (0.2%), and Mo (0.05%). Macronutrients (N, P, and K) were applied annually; micronutrients were applied once at the start of the experiment (year 1).

Exclosure treatments ("fence") were 230 cm tall. The lower 90 cm was surrounded by 1 cm woven wire mesh with a 30 cm outward-facing flange stapled to the ground to exclude digging animals (e.g., rabbits and voles), though not fully subterranean ones (e.g., gophers and moles). The upper 90 cm had three evenly spaced barbless wires to restrict larger vertebrate access (e.g., bison, elk, reindeer, or kangaroos). A few sites deviated from this fence design (Supplementary Table 2). While all sites had a wild herbivore community (e.g., a mix of rodents, lagomorphs, ungulates, marsupials, etc), domestic livestock (e.g., sheep, yak, goat, and cattle) were present at ten sites (Supplementary Table 1), allowing us to compare herbivore effects at sites with managed herds.

**Vegetation sampling.** We analyzed annual peak season live biomass by measuring aboveground biomass of all plants rooted within two 0.1 m$^2$ (10 × 100 cm) strips in each experimental plot. Clipped vegetation was separated into live and dead components, dried at 60 °C for 48 h, and weighed to the nearest 0.01 g. We collected all leaves and current year's woody growth from shrubs and subshrubs occurring in plots. We visually estimated the percent cover of each species to the nearest 1% in a randomly designated, but permanently marked, 1 × 1 m subplot within each 25 m$^2$ plot to quantify species richness and composition.

**Herbivory.** We quantified potential vertebrate herbivore impact in two ways. First, we used a published empirical metric of herbivore impact intensity and frequency ("herbivore index")[38]. In brief, all herbivore species that consume grassland biomass throughout the year were documented by the PI at each site (>2 kg), and PIs assigned an importance value for each species that reflected the impact or frequency of encounter, from 1 (present, but low impact and frequency) to 5 (high impact and frequency). An index value was calculated for each site as the sum of herbivore importance values for all herbivores[38]. This empirical herbivore index, based on a standardized rubric completed for each site, accounts for site-level variation in herbivore abundance and diversity, integrated across seasons and years. Second, we extracted the modeled terrestrial potential wild grazer biomass from a published dataset[39] ("modeled herbivore biomass") using site-level latitude and longitude values. We included the model-estimated value of herbivore biomass for each site location as a second standardized metric of potential herbivore impact among our experimental sites. While each of these provides information about potential and actual grazing intensity, neither is a direct site- or treatment-scale

measure. Finally, although studies in some grasslands have shown arthropods can control plant biomass, we did not include insect herbivory in this study because previous work in this experiment suggests that arthropods increase in biomass with increasing plant biomass, but they do not strongly suppress plant biomass in any of the treatments[55]. Thus, we focus here on the impacts of fences—and vertebrate herbivory—because evidence suggests that arthropod herbivores are impacted by the treatments, but have little overall effect on the treatments.

**Soils.** We collected two 2.5 cm diameter by 10 cm depth soil cores, free of litter and vegetation, from each plot prior to initiation of the experiment (year 0—"Y0"). We composited cores from each plot, homogenized through a 2 mm sieve, air dried, and assayed for %N and %C, using dry combustion GC analysis (COSTECH ESC 4010 Element Analyzer, University of Nebraska, Lincoln, NE, USA) and also assayed for soil phosphorus, potassium, and micronutrients, soil pH, organic matter, and texture (A&L Analytical Laboratory, Memphis, TN, USA). Because the site-scale correlation between the ambient soil %N and %C was high (0.96, $P < 0.001$), we included %N in our models.

**Climate.** We characterized site-level climate and seasonality over 10–30 year timespans using the WorldClim database (version 1.4; http://www.worldclim.org/bioclim)[56] associated to sites via latitude and longitude. We included mean annual temperature (°C; "BIO1" in the WorldClim database), mean annual precipitation (mm per year; BIO12), precipitation variability (coefficient of variation in precipitation among months; BIO15), rainfall-potential evapotranspiration (mm per month, PET data from CGIAR)[57], temperature variability (standard deviation of temperature among months; BIO4), and mean precipitation in the warmest quarter (mm; BIO18).

**Atmospheric nitrogen deposition.** We characterized site-level atmospheric nitrogen input using modeled N-deposition (kg N ha$^{-1}$ y$^{-1}$) associated with study sites via latitude and longitude. Model input included measurements and future projections using a global three-dimensional chemistry-transport model TM3 (ref. [58]). Because our study sites span continents, the 5-degree longitude by 3.75-degree latitude model and output grid resolution (50 × 50 km sub-grids) were sufficient to differentiate N-deposition rates among sites.

**Statistical analyses.** To test the hypothesis that vertebrate herbivores in grasslands can counterbalance the increased primary production due to eutrophication, we assessed the effect of each experimental treatment on the change in plot-scale biomass starting with a mixed-effects model (lmer function in the lme4 R library), with site and treatment year (number of years treatments had been applied) nested within site as random intercepts. We used the Satterthwaite's degrees of freedom method to estimate $P$ values for the mixed-effects models. Fertilization and fencing were fixed effects, and live biomass was log$_{10}$ transformed. Biomass change in response to fencing was estimated as log($B_{f+}$) − log ($B_{f−}$), where $B_{f+}$ was average site-level live biomass in fenced plots and $B_{f−}$ was average site-level live biomass in control plots. For the other treatments, we performed the analogous calculations. We examined residuals for homogeneity of variance.

To test whether treatment effects decline with time, we analyzed the dataset of 58 sites with pretreatment and 2 or more years of experimental data, as well as the subset of 42 sites with 5 or more years of data and 24 sites with 8 or more years of data. We additionally tested whether the same main effects, interactions, and time-independence were obtained with only the subset of sites with 8–10 years of data; models were qualitatively similar to the full dataset, so we do not present this analysis. Two additional desert sites had no live biomass in a substantial proportion of the plots across all sampling dates (45% of ethamc.au plots and 33% of ethass.au plots). The numerous zeros destabilized statistical models; all model results were sensitive to the statistical choices for addressing the zeros at these two sites (e.g., adding a small value). Thus, we excluded these two sites from the current analyses for statistical reasons. All analyses were performed using R (version 3.2.3; R Foundation for Statistical Computing).

To determine the conditions under which herbivory most strongly counteracted the effect of nutrients on aboveground biomass, we fit mixed-effects models that included all interactions between the fertilization, fencing, and environmental covariates. We included site-level species richness (all taxa in all plots and years from each site), species turnover estimated as mean community distance between each posttreatment year's composition and pretreatment composition (Jaccard distance), site-level N-deposition, climate factors, and plot-scale soil chemistry (soil N, soil P, and pH). We did not have a complete set of soil chemistry and N-deposition data for all sites, so we separately examined the effects of soil N, P, and pH and N-deposition in a regression that included the 36 sites for which these data were available.

**Reporting summary.** Further information on research design is available in the Nature Research Reporting Summary linked to this article.

## Data availability
Source data (plant, herbivore, and soil nitrogen) are provided with this paper. The plant, herbivore, and soil nitrogen data presented in the current study are also available in the

Environmental Data Initiative (EDI) repository[59] with the identifier https://doi.org/10.6073/pasta/a318fe0fb11eb43c1a2c8233b2e3494f. The WorldClim database (version 1.4) is available at http://www.worldclim.org/bioclim. The modeled herbivore mass data by Zhu et al.[39] are available in the PANGAEA repository at https://doi.org/10.1594/PANGAEA.884853.

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

## Acknowledgements
This work was generated using data from the Nutrient Network (http://www.nutnet.org) experiment, funded at the site scale by individual researchers. Author contributions are detailed in the "Author contributions" section and Supplementary Table 5; Supplementary Table 6 lists all data contributors who are not authors. Coordination and data management have been supported by funding to E.T.B and E.W.S. from the National Science Foundation Research Coordination Network (NSF-DEB-1042132) and Long-Term Ecological Research (NSF-DEB-1234162 and NSF-DEB-1831944 to Cedar Creek LTER) programs, and the Institute on the Environment (DG-0001-13). We also thank the Minnesota Supercomputer Institute for hosting project data and the Institute on the Environment for hosting Network meetings. The authors dedicate this paper to the memory of Enrique Chaneton who contributed data and early ideas to this work.

## Author contributions
E.T.B. and E.W.S. developed and framed the research question, analyzed the data, and coordinate the Nutrient Network collaboration. W.S.H. contributed to analyses. E.T.B., W.S.H., P.B.A., M.N.B., M.W.C., S.C., M.C.C., C.A.A., C.R.D., T.L.D., I.D., A.E., J.L.F., D.S.G., R.W.H., K.J.K., L.S.L., A.S.M., J.P.M., J.L.M., B.M., H.O.V., S.A.P., J.N.P., A.C.R., M. Sankaran, M. Schütz, C.J.S., R.V., P.A.W., and E.W.S. contributed data. E.T.B. wrote the paper and all other authors (W.S.H., P.B.A., C.A.A., M.N.B., M.W.C., M.C.C., C.R.D., T.L.D., I.D., A.E., J.L.F., P.G., D.S.G., R.W.H., A.M.K., K.J.K., L.S.L., A.S.M., J.P.M., J.L.M., B.M., R.O.H., H.O.V., S.A.P., J.N.P., A.C.R., M. Sankaran, M. Schütz, J.S., C.J.S., R.V., P.A.W., and E.W.S.) contributed to paper writing.

## Competing interests
The authors declare no competing interests.

## Additional information

E. T. Borer [1✉], W. S. Harpole [2,3,4], P. B. Adler [5], C. A. Arnillas [6], M. N. Bugalho [7], M. W. Cadotte [8], M. C. Caldeira [9], S. Campana [10], C. R. Dickman [11], T. L. Dickson [12], I. Donohue [13], A. Eskelinen [2,3,14], J. L. Firn [15], P. Graff [10], D. S. Gruner [16], R. W. Heckman [17,18], A. M. Koltz [19], K. J. Komatsu [20], L. S. Lannes [21], A. S. MacDougall [22], J. P. Martina [23], J. L. Moore [24], B. Mortensen [25], R. Ochoa-Hueso [26], H. Olde Venterink [27], S. A. Power [28], J. N. Price [29], A. C. Risch [30], M. Sankaran [31,32], M. Schütz [33], J. Sitters [27], C. J. Stevens [33], R. Virtanen [14], P. A. Wilfahrt [1,34] & E. W. Seabloom [1]

[1]Department of Ecology, Evolution and Behavior, University of Minnesota, St. Paul, MN, USA. [2]Helmholtz Center for Environmental Research – UFZ, Department of Physiological Diversity, Permoserstrasse 15, 04318 Leipzig, Germany. [3]German Centre for Integrative Biodiversity Research (iDiv), Deutscher Platz 5e, 04103 Leipzig, Germany. [4]Martin Luther University Halle-Wittenberg, am Kirchtor 1, 06108 Halle (Saale), Germany. [5]Department of Wildland Resources and the Ecology Center, Utah State University, Logan, UT, USA. [6]Department of Physical and Environmental Sciences, University of Toronto - Scarborough, Toronto, ON, Canada. [7]Centre for Applied Ecology (CEABN-InBIO), School of Agriculture, University of Lisbon, Tapada da Ajuda, Lisbon, Portugal. [8]Department of Biological Sciences, University of Toronto - Scarborough, Toronto, ON, Canada. [9]Forest Research Centre, School of Agriculture, University of Lisbon, Tapada da Ajuda, Lisbon, Portugal. [10]IFEVA, Universidad de Buenos Aires, CONICET, Facultad de Agronomía, Buenos Aires, Argentina. [11]School of Life and Environmental Sciences, The University of Sydney, Sydney, NSW, Australia. [12]Department of Biology, University of Nebraska at Omaha, Omaha, NE, USA. [13]Department of Zoology, School of Natural Sciences, Trinity College Dublin, Dublin, Ireland. [14]Department of Ecology & Genetics, University of Oulu, Oulu, Finland. [15]School of Earth, Environmental and Biological Sciences, Queensland University of Technology, Brisbane, QLD, Australia. [16]Department of Entomology, University of Maryland, College Park, MD, USA. [17]Department of Biology, University of North Carolina, Chapel Hill, NC, USA. [18]Department of Integrative Biology, University of Texas, Austin, TX, USA. [19]Department of Biology, Washington University in St. Louis, St. Louis, MO, USA. [20]Smithsonian Environmental Research Center, Edgewater, MD, USA. [21]Department of Biology and Animal Sciences, São Paulo State University - UNESP, São Paulo, Brazil. [22]Department of Integrative Biology, University of Guelph, Guelph, ON, Canada. [23]Department of Biology, Texas State University, San Marcos, TX, USA. [24]School of Biological Sciences, Monash University, Clayton Campus, Clayton, VIC, Australia. [25]Department of Biology, Benedictine College, Atchison, KS, USA. [26]Department of Biology, IVAGRO, University of Cádiz, Cádiz, Spain. [27]Department of Biology, Vrije Universiteit Brussel, Brussels, Belgium. [28]Hawkesbury Institute for the Environment, Western Sydney University, Richmond, NSW, Australia. [29]Institute of Land, Water and Society, Charles Sturt University, Albury, NSW, Australia. [30]Swiss Federal Institute for Forest, Snow and Landscape Research, Birmensdorf, Switzerland. [31]National Centre for Biological Sciences, TIFR, Bengaluru, India. [32]School of Biology, University of Leeds, Leeds, UK. [33]Lancaster Environment Centre, Lancaster University, Lancaster, UK. [34]Department of Disturbance Ecology, University of Bayreuth, Bayreuth, Germany. ✉email: borer@umn.edu

