## [Peer Review File · Nature Communications]

Reviewers' Comments:

Reviewer #1:

Remarks to the Author:

The MS 'Nutrients cause grassland biomass to outpace herbivory' uses data from the NutNet project to analyze the role of herbivory and nutrient addition, as main and interacting effects, on grassland plant biomass with the goal of testing whether additional biomass produced in fertilized plots is consumed by herbivores. They conclude that the main hypothesis is not supported, but that both nutrient supply and vertebrate herbivory can control aboveground biomass, particularly at sites with low precipitation and sites with high ambient soil nitrogen.

There is a rich body of theory and empirical study on plant-animal interactions, and the introduction touches briefly on some of the key papers. However, I think the references cited that build on trophic interaction theory tends to complicate the main thrust. So, when referring to the Polis, Oksanen, Volterra, Turkington, and Chase papers it made me think that this MS would delve into top-down, bottom-up dynamics – not simply plant-herbivore, but also predator-prey and trophic cascades. And then, with the reference to Coley's research I was also expecting the addition of plant-defense theory. I suggest a tighter focus on plant-herbivore interactions.

The study is straightforward and the results are very interesting. I'm just not entirely convinced that this is a truly novel groundbreaking finding because I think we lack a critical piece of information. There are many fenced enclosure studies in the literature. And many of these suffer the same limitation. Grazing intensity, or stocking rate, is not available. In this study, grazing intensity, I think, is a critical component because the entire premise is whether grazers respond to increased biomass as a result of fertilization. We can see from Fig 2 that biomass, overall, is greater in fenced areas, but the absolute difference is minimal ($\sim 1 \text{ g/m}^2$). In reading this MS we have to assume that wild herbivore grazing occurs in plots that have not been fenced. Is this truly the case for all sites? What is the grazing utilization rate season-to-season and year-to-year? How accessible are each site to wild herbivores and what evidence is there that grazing normally occurs in the vicinity of each site?

Reviewer #2:

Remarks to the Author:

See attachment.

Reviewer #3:

Remarks to the Author:

The Borer et al. manuscript represents a substantial study across numerous grassland/meadow sites where responses to exclusion of vertebrate herbivores and addition of nutrients, alone and in combination, are compared to controls. The meta-analysis conclusion is that with nutrient addition vertebrate herbivores cannot consume all of the additional plant production, but consume a constant proportion of the production with or without nutrient addition, and nutrient addition and herbivory are additive effects with nutrient addition being more important. Furthermore, the strength of the responses is compared over study duration, site original soil nutrient content, and site precipitation, finding that study duration had no effect, and responses were greater at low precipitation and high original soil nutrient sites.

The authors claim that "theory" predicts that vertebrate herbivores should be able to consume all additional primary production with added nutrients, and their study disproves the "theory". However, this is just one theoretical construct. Long-established alternative theories (e.g., Arditi & Ginzburg 1989, Liebold 1989, Schmitz 1992) predict that herbivores consume a constant

proportion of plant production with increased primary production, as reported in this study. Therefore, the authors have failed to adequately represent the ecological literature, and claim theoretical novelty, which is not the case.

While the authors refer to the contextual nature of the responses in their various sites, a contextual framework raises numerous questions to me that the authors ignore:

- Do individual sites all respond the same or do some sites fail to respond in accordance with the meta-analysis outcome, and if so, why?
- What are the numerical responses of vertebrate herbivores to increased primary production and the time scale needed to see this in the experiments?
- Do sites differ in the abundance and role of predators, and if so how might this affect the meta-analysis?
- The presence of domestic vertebrate herbivores implies management and the authors assume that the presence of wild vertebrate herbivores alone implies the absence of management, but hunting, habitat manipulation, etc. are forms of management for wildlife; how may this impact results?
- Vertebrate herbivores are not the only herbivores (e.g., insects); do these other herbivores affect nutrient enhanced primary production in the same manner as suggested by the meta-analysis and what is the cumulative herbivory response?

Granted the length limits of the journal cramp full discussion of the breadth of ecological theories, as well as the above and other contextual complexities, but they cannot be ignored, and the meta-analysis result cannot be presented as a new or constant result. In fact, the manuscript's discussion section is very weak. Finally, the authors claim that their findings are important to fire management, food security and climate change with no discussion of how or why.

I have a few editorial comments.

- Line 91. Eutrophied? Nutritious, not nutritional.
- Line 201 – 202. "Past studies have found conflicting effects of herbivores on biomass, including strongly increasing 42,43 or decreasing 43-45 biomass." This sentence makes no sense in a paragraph dealing with species richness and turnover.

Schmitz, O. J. (1992). "Exploitation in model food chains with mechanistic consumer-resource dynamics." *Theoretical Population Biology* 41(2): 161-183.

Arditi, R. and L. R. Ginzburg (1989). "Coupling in predator-prey dynamics: ratio-dependence." *Journal of Theoretical Biology* 139: 311-326.

Leibold, M. A. (1989). "Resource edibility and the effects of predators and productivity on the outcome of trophic interactions." *American Naturalist* 134: 922-949.

REVIEWERS' COMMENTS:

Reviewer #1 (Remarks to the Author):

The MS 'Nutrients cause grassland biomass to outpace herbivory' uses data from the NutNet project to analyze the role of herbivory and nutrient addition, as main and interacting effects, on grassland plant biomass with the goal of testing whether additional biomass produced in fertilized plots is consumed by herbivores. They conclude that the main hypothesis is not supported, but that both nutrient supply and vertebrate herbivory can control aboveground biomass, particularly at sites with low precipitation and sites with high ambient soil nitrogen.

There is a rich body of theory and empirical study on plant-animal interactions, and the introduction touches briefly on some of the key papers. However, I think the references cited that build on trophic interaction theory tends to complicate the main thrust. So, when referring to the Polis, Oksanen, Volterra, Turkington, and Chase papers it made me think that this MS would delve into top-down, bottom-up dynamics – not simply plant-herbivore, but also predator-prey and trophic cascades. And then, with the reference to Coley's research I was also expecting the addition of plant-defense theory. I suggest a tighter focus on plant-herbivore interactions.

##We definitely agree about the rich body of trophic theory: it is difficult to pick the best papers to cite. The papers we cite were chosen carefully and intentionally, though. In particular, we cited the Volterra paper because section 3 of that paper lays out the origin of dynamic predictions for cases "when one species feeds upon another." Although it has come to be called 'predator-prey theory,' this was not Volterra's original terminology. While it is often cited in the trophic cascade literature, the Oksanen et al paper lays out predictions for plant-only and plant-herbivore systems. By setting the herbivores to zero in the range of productivity where herbivores can persist in the system (analogous to putting up a fence), this theory also provides predictions for plant biomass in the presence and absence of herbivores across the same range of productivity. We show these scenarios in Fig 1a of our paper.

##While Turkington and Chase are known for their work that includes predators, the papers we cite are focused on plant-herbivore dynamics. Chase et al (2000) lay out dynamic predictions for plant-herbivore interactions across a productivity gradient and examine the evidence with meta-analysis. While predators are considered in two paragraphs of the paper, the focus of that work is on the plant-herbivore interactions as a function of environmental productivity. In his paper, Turkington specifically identifies a gap and calls for a plant-vertebrate herbivore experiment such as the one we present. Because these papers have key elements that are directly relevant to the work of this paper, we have retained them.

##To increase the focus on theory about plant-herbivore interactions, we now cite Crawley's 1983 book (*Herbivory: the dynamics of animal-plant interactions*) in which he develops a 2-level plant-herbivore model (with logistic plant growth). He does not focus on productivity gradients, though, whereas the core of each of the Oksanen and Chase models is a productivity gradient along which differing predictions for plant biomass emerge. In each of these models, the range of the model where plants and herbivores are present, with increasing productivity, herbivores have an increasing impact on observed plant mass.

##We have removed some of the citations as suggested. We cited Polis' beautiful Oikos essay on why the world is green as support for a statement about climate being a major driver, but we have removed this citation because the meta-analysis by DelGrosso et al is probably sufficient to make the point. We also cited Polis and Strong's Am Nat paper that specifically considers the dynamic

consequences of food web complexity, including edibility. However, R3 recommended citing Leibold 1989, so we have removed both the Murdoch and the Polis and Strong reference as somewhat redundant.

##Finally, while it was not a theory paper, we cited the meta-analysis by Endara & Coley because a key result in that paper was that high resource environments tend to be dominated by undefended plants, suggesting that these environments may have the greatest potential for herbivore control of plant biomass. We have revised the text to highlight this implication of their results.

The study is straightforward and the results are very interesting. I'm just not entirely convinced that this is a truly novel groundbreaking finding because I think we lack a critical piece of information. There are many fenced enclosure studies in the literature. And many of these suffer the same limitation. Grazing intensity, or stocking rate, is not available. In this study, grazing intensity, I think, is a critical component because the entire premise is whether grazers respond to increased biomass as a result of fertilization. We can see from Fig 2 that biomass, overall, is greater in fenced areas, but the absolute difference is minimal (~1 g/m²). In reading this MS we have to assume that wild herbivore grazing occurs in plots that have not been fenced. Is this truly the case for all sites? What is the grazing utilization rate season-to-season and year-to-year? How accessible are each site to wild herbivores and what evidence is there that grazing normally occurs in the vicinity of each site?

##This is a very good point, and one we have worked hard to address. We now include two different quantifications of herbivore impact. First, we include a metric of grazing intensity published by Anderson et al. 2018 in Ecology. This empirically-derived "herbivore index" metric is based on a standardized rubric used by all PIs in this project. This metric accounts for site-level variation in herbivore abundance and diversity, integrated through a year, providing a comparable metric of herbivore impact intensity and frequency across the sites in this experiment. Second, we include modeled herbivore biomass data, extracted from a recent paper by Zhu et al 2018 that modeled global potential terrestrial grazer biomass. While this biomass metric more clearly addresses a point by reviewer 2, it provides a second quantification of potential consumer impact.

##When we included these metrics in our models examining covariates, we found that the modeled grazer biomass was never a significant predictor. However, the herbivore index was not only significant, it also supported our original analyses of the sites with livestock by demonstrating that all of the additional fertilized plant biomass is removed only where grazing intensity is very high. At most sites with low to intermediate consumption intensity, herbivores do not keep up with the additional fertilization-induced biomass.

##We have added a figure (new Fig. 3a) and text in the main document, text in the methods, and updated the table of model results in the extended data section.

Anderson et al 2018 Ecology 99: 822-831. doi: 10.1002/ecy.2175

Zhu et al. 2018 Nat Ecol Evol 2018. 2(4): 640-649. doi:10.1038/s41559-018-0481-y

Reviewer #2 (Remarks to the Author):

Key results: Borer et al use a global dataset of manipulations of both herbivory and nutrient availability to test whether herbivores can remove additional biomass production due to

eutrophication. Second, they test under whether these effects are context-dependent by linking them to gradients of soil nutrients and climate. They find that herbivore consume about 12% on average of the increased biomass following nutrient addition. Importantly, the interaction was not significant suggesting the herbivore effect is proportional to the grazing intensity under “natural” conditions. Furthermore, herbivore effects are largest at drier and nutrient rich sites.

Validity: I do not believe there are major flaws in the study that should prevent publication.

##We appreciate these positive remarks.

However, there a some important assumption being made which should be discussed in more detail. 1) The use of permanent exclosures is known to underestimate both primary productivity and grazing intensity (see McNaughton et al (1996) Ecology). This could have underestimated the herbivore effect, although it is hard to predict the size of this bias. I understand the logistical difficulties in executing a large scale experiment, and that might have prevented them from using for example movable exclosures. Nevertheless, this should be discussed including the possible consequences for the results/conclusions.

##The reviewer is correct that we do not have strong measures of productivity, and that moveable cages are one way to empirically estimate productivity in the absence of mammalian grazers. However, the theory we examine in this paper predicts standing biomass, not productivity, so we believe that we have empirically captured the correct response variable to test these predictions. We were careful not to make inference about primary productivity (rate of carbon capture or plant growth rate), but focused our inference on plant biomass (standing stock).

##Nonetheless, the point about grazing intensity is a good one, and in this version, we have added site-level empirical data on grazing intensity as well as modeled herbivore mass, which we believe addresses this and later points. We provided a more extensive explanation of our changes in response to this concern in our reply to R1.

2) The small-scale application of nutrients is different from landscape-wide eutrophication. This could either increase or decrease the measurements of herbivore consumption. This is hardly discussed, yet is at the core of their rationale.

##This is also a good point. While we cannot directly scale these results up, we do observe in the current version that “long-term nutrient deposition has led to increased live plant biomass in the presence of herbivores across a wide range of global grasslands” – suggesting that the weak effect of herbivores observed in our experimental plots is consistent with observations of the impact of widespread grassland eutrophication at landscape scales.

3) The biomass measurements were taken at peak season standing biomass. However, herbivores also forage during the remaining part of the year. Again, this might have underestimated the herbivore effect and should be discussed.

##This is true, and we believe that our new analyses account for this by including a metric of site-level grazing intensity that is integrated through a year. In particular, the empirically-derived grazing intensity metric we include in the current version of the manuscript accounts for site-level variation in herbivore abundance and diversity, and provides a comparable metric of herbivory impact intensity and frequency through a year at each site. In addition, since we seek to understand herbivore effects on plant mass (not the rate of productivity), peak season measurements of the plants provide information about the maximum annual mass.

Originality and significance: This is the weakest part of the study. The authors emphasize several times that there are no large-scale analyses of combined herbivore and nutrient manipulations and this is a key research gap. However, a study of Borer et al (2014) Nature, which shares many of the same authors (including the lead author) has presented exactly such analysis. Fig. 2B of that paper is almost identical to Fig. 2 of the current manuscript. The fact that this is not mentioned or referred to is quite remarkable.

##We can see why the reviewer sees similarities with the 2014 figure. Fig 2b from the 2014 paper is a graph of total biomass (live + dead), not just live biomass, as in the current study. Our analysis in the earlier study quantified the degree to which total biomass was an effective proxy for light interception. Because dead biomass varies enormously among sites but is broadly correlated with live mass, the summary data appear similar. However, the combination of live and dead biomass data presented as a covariate in the 2014 paper are not an appropriate response variable to test herbivore effects on their resources, or live plant mass, that we present, here. The earlier paper's focus was to determine the impacts of fencing and nutrients on grassland biodiversity, and we found that light interception (not total biomass) provided a strong explanation for variation in biodiversity loss. Since the questions and response variables differ between these two papers, we did not feel that the earlier work was relevant to cite for the current questions. The current work is the first NutNet paper to analyze the impacts of herbivores on live mass. If the reviewer and editor feel that this earlier paper should be more extensively discussed, we can find a way to include it.

Furthermore, because the interaction between fencing and nutrients is not significant (as already presented in the Borer et al (2014) Nature paper), the remaining part of the manuscript is very comparable to grazing intensities without nutrient addition. Over the last decades there has been an extensive amount of work that investigates drivers of herbivore consumption across large-scale gradients including a meta-analysis of 236 experiments by Milchunas and Lauenroth (1993) Ecological Monographs. Also, work by East (1984) Afr. J. Ecol. has investigated the relationships between herbivore biomass densities, rainfall and soil nutrients. Herbivore biomass densities are key in explaining the herbivore grazing intensities, but seems to be missing completely. As such, I do not think the results presented are highly original. Neither do I think the manuscript will influence thinking of scientist in this field.

The work we present here builds from excellent meta-analyses focused on predicting plant biomass (e.g. Milchunas & Lauenroth, Gruner et al 2008) to experimentally test – using identically replicated experimental methods and sampling – for the theoretically predicted interaction between fertilization and grazing. Milchunas and Lauenroth foreshadowed this with observed covariates, but Gruner et al (2008 Ecology Letters 11:740-755) demonstrated that there was insufficient data to test for this. An interaction test is most effectively done with an experiment, and determining the generality of the test requires an experiment replicated under a wide variety of conditions. Our focus was on testing for the interaction between fertilization and fencing, so we had not cited Milchunas and Lauenroth, but we have now added this because of this paper's importance for generating expectations for the fencing effects on live plant biomass. The East meta-analysis is another excellent data compilation study, but it focuses on predicting herbivore biomass from rainfall and soils, thus has a different focal response variable, so we have not included this. However, we now include both grazing intensity and modeled herbivore biomass (modeled from relationships like those examined in East '84) as explanatory variables.

##Although the overall interaction is not significant, we use our experiment to build from these earlier meta-analytical studies to demonstrate that the interaction between herbivores and nutrients IS important in low, not high, precipitation sites, high resource environments, and sites with high grazing intensity. Further, we believe that because it is a replicated experiment, the test

of this hypothesis is strong and the result – that the interaction is not significant globally but is important under certain conditions – is both important and consistent with e.g., impacts of N-deposition in the presence of herbivores. We hope that our revised version of this manuscript clarifies how this work builds from previous work and provides new insights into the context-dependence of the role of herbivores and nutrients in controlling grassland plant mass.

Data & methodology: The study builds on an experiment across 58 sites, which yields a fantastic dataset to test general patterns. Except from the validity points addressed above and some minor comments below, I do not have any specific comments.

##We appreciate the kind words about the project!

Appropriate use of statistics and treatment of uncertainties: The statistics and presentation of the results should be clarified at some points (see specific comments below). In general the statistics and presentation seem appropriate.

##We address these, below.

Conclusions: The conclusions seem robust and reliable.

##We are glad the reviewer felt that the conclusions are robust and reliable.

Suggested improvements: The key parameter that is missing in this study is herbivore biomass density. I expect it would explain the effects across environmental gradients, the differences between livestock and wildlife and present the most direct predictor of herbivore effects. Furthermore, it would strengthen the link to the “green world hypothesis” that serves as the theoretical underpinning of the manuscript. If the authors could add data on herbivore biomass densities, this would greatly benefit the manuscript.

##As we described, above, we have now added two different measures of potential site-level herbivore impact: modeled herbivore biomass and a metric of grazing intensity. The modeled biomass was based on factors shown to be important in East et al 1984 – soils and meteorological variables – which determine the food that constrains herbivore population size, and is used directly to estimate herbivore mass. Grazing intensity published by Anderson et al. (2018 Ecology) is an empirically-derived metric based on a standardized rubric used by all PIs across the NutNet project.

##Interestingly, and surprisingly to us given the variation in the fence x nutrient interaction across gradients of soils and precipitation, the modeled herbivore biomass data was never a significant predictor. However, the grain of these modeled data is fairly coarse and based on proxy variables, and represents potential herbivore mass, rather than realized mass (or impact). The site-level grazing intensity data, though, served as an important covariate. Interestingly, though, this index varied fairly independently of soils, precipitation, and even domestic livestock, so was not a better predictor of the variation in the strength of the fence x nutrient interaction. In short, we absolutely agree that these additional herbivore data strengthened the paper, but the abiotic gradients remain as significant factors in the models, even with the additional variance explained by the herbivory intensity. For us, it was exciting to see how the link to the “green world hypothesis” that these data provided served to strengthen and enrich the key results of our experiment.

References: References to studies on drivers of grazing intensity and herbivore biomass densities should be included. For example:

Milchunas, D.G. and Lauenroth, W.K. (1993), Quantitative Effects of Grazing on Vegetation and Soils Over a Global Range of Environments. *Ecological Monographs*, 63: 327-366. doi:[10.2307/2937150](https://doi.org/10.2307/2937150)

East, R. Rainfall, nutrient status and biomass of large African savannah mammals. *Afr. J. Ecol.* 22, 245± 270 (1984).

McNaughton, S. J., Oesterheld, M., Frank, D. A. & Williams, K. J. Ecosystem-level patterns of primary productivity and herbivory in terrestrial habitats. *Nature* 341, 142±144 (1989).

McNaughton, S. J. Ecology of a grazing ecosystem - the Serengeti. *Ecol. Monogr.* 55, 259–294 (1985).

Hempson, G. P. *et al.* Ecology of grazing lawns in Africa. *Biol. Rev.* 90, 979–994 (2014).

Hempson, G. P., Archibald, S. & Bond, W. J. A continent-wide assessment of the form and intensity of large mammal herbivory in Africa. *Science (80-)*. 350, 1056–1061 (2015).

McNaughton, S. J., Milchunas, D. G. & Frank, D. A. How can net primary productivity be measured in grazing ecosystems? *Ecology* 77, 974–977 (1996).

Clarity and context: The manuscript is clearly written, although some parts of the methods and results section could be further clarified. See below.

Specific comments:

Line 58: Why do the authors expect herbivores control will decline when regions become more eutrophic? The previous sentence concludes that herbivores most effectively reduced biomass at sites with high soil N.

##We appreciate this comment and have modified the text to focus on our experimental results rather than the observed covariates, “Overall, these experimental results suggest that grassland biomass will outstrip wild herbivore control as human activities increase elemental nutrient supply, with widespread consequences for grazing and fire risk.”

Line 59: The link with biodiversity management is unclear and seems to come out of the blue.

##We have removed this statement from the current abstract.

Lines 79-86: I am surprised the authors do not cite the Borer et al (2014) Nature paper which shares many of the authors. This study clearly shows that nutrients had a large effect on biomass and herbivores had a small (and contrasting) effect.

##We hope that our explanation above, that while these figures appear the same, they represent different response variables (live biomass that is consumed (or not) by herbivores vs total (live + dead) biomass), has clarified why we have not discussed this earlier paper. If the reviewer and editor feel we should more extensively discuss this paper, we can do this.

Lines 87-99: Although not many studies have examined grazing intensities with and without manipulation of nutrient availability, there is a large number of studies that examined drivers of grazing intensity across large environmental gradients. A key indicator is herbivore biomass density, which seem to be missing in the introduction (and the study as a whole).

##We have added two metrics of herbivore density to the current analyses, and accordingly revised the conceptual framing in the introduction in response to reviewer comments, as described more fully above.

Line 115: I think for clarity “terrestrial” should be added to this sentence.

##Done.

Lines 178-181: This really depends on how you define control. At low precipitation herbivores will remove a proportion of biomass/primary production. This is the focus of the current manuscript. If the authors would have investigated the absolute consumption, the patterns would be expected to be opposite. This could be further clarified.

Proportional removal implies increased absolute removal, so yes, this implies that more total biomass is being removed at more productive sites. Our results demonstrate that the net effect of the consumer food web leads to a proportional removal that is constant across the precipitation gradient. If this were caused by proportional predation rates changing, the proportional change would not remain constant. To clarify the predictions from theory, we have changed this text to, “...these results do not support the hypothesis of an increasing rate of predator control of herbivores...” In addition, with a few notable exceptions (e.g. the site in Tanzania), large carnivores are typically quite rare in most current day grassland ecosystems, so with respect to large herbivores, many grasslands function as 2-level systems.

Lines 183-187: Isn't this part of the discussion? There are no measurement on stoichiometry presented in the manuscript.

##We have added the citation to the work that shows this pattern in this experiment (Anderson et al 2018). We add that in the current paper, we have combined the Results and Discussion within one section, as indicated by the new header for this section.

Lines 189-190: This is an important limitation of the study that could have affected the conclusions and should be more extensively discussed (preferably in the discussion).

Our current study is set up to ask whether current herbivore densities can keep up with consumption of the additional biomass produced under fertilized conditions. In terms of this specific result, the only conditions in which the difference between preference vs pop dynamics might modify the conclusions would be in the driest sites, so the overall result of proportional consumption in grasslands is unlikely to be changed.

##We have modified the text to place this dry site result into a landscape-scale context, “While population dynamics of herbivores in some low precipitation systems can vary with the quality of their resources, elevated nitrogen inputs can lead to increased standing biomass even in dryland systems, suggesting that this preference response in low precipitation regions may be weaker at the landscape scale.”

Lines 196-206: In this discussion on dominance, the recent paper by Koerner et al (2018) in NEE seems an appropriate reference.

##We have added this reference.

Lines 208-209: I disagree. See Borer et al (2014) Nature. Or Michulnas et al (1993) Ecological Monographs.

##Thanks for pointing this out. We realized that this was phrased too generally, and we have revised the text to better capture the novelty of the current work. In particular, we have rephrased this to more clearly restate the core conclusion of the Turkington (2009) article that specifically addressed this: “This work fills a key knowledge gap: previous syntheses have bemoaned the scarcity of manipulative experimental studies testing hypotheses about the generality of effects of vertebrate herbivores and nutrient supply on terrestrial biomass production.”

Lines 221-223: This is a pretty bold statement that is not supported by the results from this manuscript, neither by references to other studies.

##At the end of the first introduction paragraph, we cite literature relating standing grassland biomass to grazing, biodiversity loss, fuel load, and fire severity. If the reviewer and editor feel we should cite this literature again in this final sentence, we can add this.

Fig.1: The use of C for both ambient nutrient levels and the presence of herbivores is confusing. Why are the C and F terms needed for herbivory? Can it not just be + and –H? Either change or better explain.

##With this comment, we now see why the labeling on this figure was confusing. We have changed to “fence” and “open” for the line labels and “control” and “nutrient” for the x-axis labels. We have also changed the letters on c, f, and i to match fig 2. They are now “control” “nutrient” and “nut*fc.” In response to some confusion about the response variable, we now label it “live plant biomass” for clarity.

Fig. 1c,f,i and 2: I believe these figures are needlessly complex. As I understand it correctly, the figures shows LogResponseRatios, but the y-axis label suggests something else (LRR do not have units).

##This is an interesting point. Log response ratios have units in log space (difference in logs, which has units – in this case, $g\ m^{-2}$). Although our models are generating a difference of two biomass values in log space (i.e. effect size), computationally, this is the same as a log ratio in untransformed space. Further, we realize that we used the term “log ratio” in the methods but our presentation here is the difference of the logged biomass. While these are mathematically equivalent, we realize that this added confusion. We now only talk about the logged difference.

##To clarify that this analysis is different from a meta-analytical log ratio approach, we have changed the methods text to read, “Biomass change in response to fencing was estimated as $\log(B_{f+}) - \log(B_{f-})$, where B_{f+} was average site-level live biomass in fenced plots and B_{f-} was average site-level live biomass in control plots.”

Furthermore, the NxF treatment uses a different “control” to determine the effect which makes it very hard to understand. At some point I thought the y-axis shows the model estimates of the three parameters from the mixed-model, which makes sense, but I don’t think this is what it represents. Why don’t the authors not just show the LRR with respect to the control treatment? And then together with the table of model outcome the results should be quite clear in my opinion. As the figures are now, it suggests that the NxF treatment has the same biomass as the control treatment. Please clarify.

##Yes, this is correct. These are the parameters from the mixed effects model. We have clarified in the Figure 2 legend that a value of zero for the interaction means no interaction, as in the model. We have chosen this approach for two reasons. First, our primary hypothesis test is based on a test for additivity of the fence and fertilization treatments on plant biomass, so this provides the most

direct visualization of this test. Second, because of the hierarchical nature of the data, with a lot of among-site variation, it is not straightforward to generate appropriate error bars for a visualization of the treatment means. We have corresponded with experts about this (e.g., Dr. Ben Bolker), and based on this consultation have opted for the current approach which more closely maps onto the underlying models and error structure.

Fig. 3: Is this a log-response ratio (as suggested in statistical analysis part of the methods sections) or absolute changes (as suggested by the y-axis label)? Please clarify.

##These panels present the (logged) difference between each treatment and the unfertilized, unfenced control at each site. By providing this visualization, it is possible to see each of the treatment effects across each of the y-axis covariates. We have changed the text in the statistical methods section, as above, to clarify that this is a difference.

Fig. 3: There are quite some sites with negative biomass changes, especially for the fenced treatment. Some sites even have substantial negative changes (>0.25). Is this just measurement error? Or are there other explanations?

##Because this is averaged across years, it is unlikely to be simple measurement error. There are many biological reasons that excluding herbivores might reduce live biomass. For example, herbivores may stimulate plant regrowth or increase nutrient cycling rates as predicted and demonstrated in the grazing lawn literature. All of these processes could reduce the relative amount of biomass inside of fences.

Also, 20 out of 58 sites (1/3) have negative values, but in figure 2 the 95% confidence intervals don't even cross the 0 line. That seems odd.

The mixed effects model visualized in Fig 2 is better at accounting for the variation among sites and years and isolating the effect of fencing from other sources of variation. In addition, the fence effects shown in Fig 2 leverage all of the fence treatments, not just the unfertilized fenced plots. All the treatment effects in Extended Data Fig 2 show the mean effects of the treatments for every site.

Line 411: Why peak season? Herbivores also eat vegetation in the remaining part of the season?

##Peak season biomass provides an estimate of the max seasonal biomass at a site. This provides a comparable point in the season for all sites around the world.

Line 449: But those results are presented in the Extended Data section right?

##We do not currently present these results. The results presented in the Extended data are models of all data for each duration, such that there are more sites included in the models of 2-4 years than those of e.g., 8-10 years. While, as this part of the manuscript describes, we repeated these analyses with just the 24 sites for which we had data for all durations (2-4, 5-7, 8-10 years) to reassure ourselves that none of our results were due to the inclusion of different subsets of sites in each duration period, it seemed like overkill (and potentially confusing) to present these three additional model tables from analyses of just the 24 sites that showed qualitatively similar results with slightly different parameter values. If the reviewer and editor feel that these tables are important to include, we can add them.

Line 458-464: It is quite hard to believe fencing did not result in a shift in community composition, especially at sites with experiments that ran for >8 years. In all herbivore enclosures I have seen so

far myself, community shift are very evident. Can the authors add material to show such analysis to convince the readers?

We used temporal turnover as a covariate to help explain variation in fence effects in this study; we did not examine turnover as a response variable. Thus, while temporal turnover occurred in response to fences (a response being examined in detail in other analyses), the temporal turnover of plant species in response to exclosures was not associated with the magnitude of the fencing effects. This likely occurred because of different plant species pools and diversity at different sites leading to different magnitude of turnover with fencing; however, a thorough analysis of the determinants and magnitude of species turnover among sites is beyond the scope of the current study. To clarify, we have changed the text to read, "In initial model runs, species turnover was never a significant or important term for explaining the direction or magnitude of fence effects"

Paragraph starting at line 456: What were the random effects in this model and did the authors include both random intercept and slope? Was there any model selection procedures on the random effects? Any model selection on the fixed effects? We really need more information here.

##The random effects are listed in each extended data table. They were site and treatment year within site. The models were just estimating random intercepts. Models that estimated different slopes for every site and year combination tended to be unstable and did not generally converge. To address this, we have clarified that we estimate these as random intercepts and have added the model formula to every extended data table.

Extended Data Table 4A: The number of degrees of freedom for the interactions seem way to high. How can there be 2516 degrees of freedom when you compare 58 sites? Replication within sites is pseudo replication which should have been corrected for by the random effects.

Extended Data Table 4B: Same as for Table 4A. Explain the degrees of freedom and model selection procedures.

There are many degrees of freedom because we have samples collected at each site across many years and plots. We have set up the random effects structure to account for the nesting of plots and treatment years within site. In order to get correct p-values for these tests, we used Satterthwaite's degrees of freedom method for the mixed effects models. We added text to the methods to clarify this, "We used the Satterthwaite's degrees of freedom method to estimate p-values for the mixed-effects models."

Extended Data Figure 1: What are the units of live mass (Y-axis)? Why are there numbers for the first three treatments, but not for the control?

##Thanks for pointing this out. We have fixed this figure to have a corrected legend and y-axis label.

Reviewer #3 (Remarks to the Author):

The Borer et al. manuscript represents a substantial study across numerous grassland/meadow sites where responses to exclusion of vertebrate herbivores and addition of nutrients, alone and in

combination, are compared to controls. The meta-analysis conclusion is that with nutrient addition vertebrate herbivores cannot consume all of the additional plant production, but consume a constant proportion of the production with or without nutrient addition, and nutrient addition and herbivory are additive effects with nutrient addition being more important. Furthermore, the strength of the responses is compared over study duration, site original soil nutrient content, and site precipitation, finding that study duration had no effect, and responses were greater at low precipitation and high original soil nutrient sites.

The authors claim that “theory” predicts that vertebrate herbivores should be able to consume all additional primary production with added nutrients, and their study disproves the “theory”. However, this is just one theoretical construct. Long-established alternative theories (e.g., Arditì & Ginzburg 1989, Liebold 1989, Schmitz 1992) predict that herbivores consume a constant proportion of plant production with increased primary production, as reported in this study. Therefore, the authors have failed to adequately represent the ecological literature, and claim theoretical novelty, which is not the case.

##We appreciate the suggestion to more explicitly talk about theory that underpins hypotheses about proportional consumption. This is the theory that underpins the scenario in Fig 1d of our paper, but we see from this comment that we had not written about it as explicitly as we should have. We have shifted the introduction to present these theories of resource-control in a more parallel structure, particularly with revisions of the third paragraph of the main text.

##It is important to note that the work we have done is not a meta-analysis. Our work is a replicated experiment. By doing a replicated experiment distributed across sites around the world, we are able to directly compare results among sites because the treatments are identical, unlike meta-analyses. Because of the distribution of biomass among sites, we examine the difference between treatment and control in log space, which is computationally the same as a log ratio, used in meta-analysis. However, by using a replicated experiment, we can also track and directly compare a wide range of covariates among sites (e.g., soil chemistry, plant richness, climate, etc) with the confidence that the predictor and response variables are not collected in different ways among sites, as in meta-analyses. Thus, while meta-analyses like Chase et al (2000), Milchunas and Lauenroth (1993) and others provide important evidence, a multi-site replicated experiment provides a test of the correlations (with e.g. herbivory, productivity) found in meta-analyses.

While the authors refer to the contextual nature of the responses in their various sites, a contextual framework raises numerous questions to me that the authors ignore:

- Do individual sites all respond the same or do some sites fail to respond in accordance with the meta-analysis outcome, and if so, why?

##There is variation in the responses to the experimental treatments among sites which is why the estimated means in Figure 2 have error bars. In this distributed, replicated experiment, each site represents a replicate, so the among-site responses are the data used to estimate the mean response. The response to the experimental treatments at each site is shown in Extended Data Figure 2. The current manuscript includes the sentence, “In spite of a very wide range of biomass responses among sites – including sites with wild herbivores where fencing more than doubled biomass (Extended Data Fig. 2) ...”

##We then use this among-site variation to examine hypotheses for among-site variation in biotic and abiotic site characteristics, which is presented in Figure 3. This final analysis demonstrates that

the impact of herbivores varies primarily along site-level gradients of soil chemistry, precipitation, and site-level variation in herbivore abundance and diversity, integrated through a year.

- What are the numerical responses of vertebrate herbivores to increased primary production and the time scale needed to see this in the experiments?

##Our experimental treatments are not applied at a spatial scale that would substantially change the dynamics of large vertebrate herbivores. Nonetheless, our quantification of the plant biomass response across a decade suggests that the plant response to each treatment is consistent for up to a decade; herbivore effects do not e.g., get stronger (or weaker) through time with elevated productivity. While we primarily draw on theory that includes dynamic herbivores, theory that assumes a static herbivore density across different resource conditions also predicts sub- and super-additivity between herbivores and nutrients (Gruner et al 2008 Ecology Letters 11:740-755).

- Do sites differ in the abundance and role of predators, and if so how might this affect the meta-analysis?

With a few notable exceptions (e.g. the site in Tanzania), large carnivores are typically quite rare in most current day grassland ecosystems, so with respect to large herbivores, many grasslands function as 2-level systems. Further, while there is certainly variation in the role of predators across the grasslands represented in this replicated experiment, the primary goal of this work was to examine the net impact of herbivores on plant biomass regardless of the factors determining the herbivore community.

- The presence of domestic vertebrate herbivores implies management and the authors assume that the presence of wild vertebrate herbivores alone implies the absence of management, but hunting, habitat manipulation, etc. are forms of management for wildlife; how may this impact results?

##The goal of this work was to examine the responses of plant biomass to experimental treatments across a wide range of sites. These sites have a wide range of herbivore densities for many reasons, and we have incorporated two different metrics of herbivore abundance and mass into the current analyses. While herbivore and habitat management is certainly interesting and important, we did not manipulate the types of herbivore management, so this is beyond the scope of inference from this study.

- Vertebrate herbivores are not the only herbivores (e.g., insects); do these other herbivores affect nutrient enhanced primary production in the same manner as suggested by the meta-analysis and what is the cumulative herbivory response?

##While individual studies have found strong effects of arthropods on plant mass – that, in some cases, interact with vertebrate herbivory – previous work in this experiment suggests that while arthropods increase in biomass with increasing plant biomass; they do not strongly suppress plant biomass (Lind et al 2017 Ecology 98: 3022–3033). We have added the sentence, “Although studies in some grasslands have shown arthropods can control plant biomass, we did not include insect herbivory in this study because previous work in this experiment suggests that arthropods increase in

biomass with increasing plant biomass, but they do not strongly suppress plant biomass in any of the treatments (Lind et al 2017)” **to the Methods section under the Herbivory methods section.**

##In addition, that previous study demonstrated that fences have little consequence for arthropod biomass; arthropod mass does not interact with the fence treatments. So, overall, arthropod mass and abundance are positively, not negatively, related to plant production; there is little evidence of widespread a suppressive impact by arthropods on plant mass.

Granted the length limits of the journal cramp full discussion of the breadth of ecological theories, as well as the above and other contextual complexities, but they cannot be ignored, and the meta-analysis result cannot be presented as a new or constant result. In fact, the manuscript’s discussion section is very weak. Finally, the authors claim that their findings are important to fire management, food security and climate change with no discussion of how or why.

##We have revised the introduction to more explicitly talk about the resource-controlled theory, and these predictions are also presented in our conceptual figure (Fig 1). We present the biotic and abiotic covariates associated with variation in the strength of the experimental responses among sites (Fig. 3), a analysis specifically focused on understanding the variance, beyond the mean response presented in Fig. 2. We have added a discussion paragraph about the cases supporting the consumer control hypothesis, but overall, the discussion in which we place our results into the context of the literature is integrated into each of the results paragraphs. We have now more clearly indicated this integration with a “Results and Discussion” header. Finally, we have removed “food security” which was intended to refer to challenges with grazing lands, but we see from this reviewer comment that our reference was unclear. At the end of the first introduction paragraph, we cite literature relating aboveground grassland biomass to grazing, biodiversity loss, fuel load, and fire severity. If the reviewer and editor feel we should cite this literature again in this final sentence, we can add this.

I have a few editorial comments.

- Line 91. Eutrophied? Nutritious, not nutritional.

##We have changed this to “eutrophic” and “nutritious”

- Line 201 – 202. “Past studies have found conflicting effects of herbivores on biomass, including strongly increasing 42,43 or decreasing 43-45 biomass.” This sentence makes no sense in a paragraph dealing with species richness and turnover.

##We have altered this section to tie this to turnover, which was our original intention. It now reads, “Past studies have found conflicting effects of herbivores on biomass, including strongly increasing or decreasing biomass, possibly due to variable responses by the species present at the site.”

Schmitz, O. J. (1992). "Exploitation in model food chains with mechanistic consumer-resource dynamics." *Theoretical Population Biology* 41(2): 161-183.

Arditi, R. and L. R. Ginzburg (1989). "Coupling in predator-prey dynamics: ratio-dependence." *Journal of Theoretical Biology* 139: 311-326.

Leibold, M. A. (1989). "Resource edibility and the effects of predators and productivity on the outcome of trophic interactions." *American Naturalist* 134: 922-949.

Reviewers' Comments:

Reviewer #1:

Remarks to the Author:

I commend the authors for their thorough work in addressing all reviewer comments. The trickiest issue was addressing estimates of grazing intensity. I am satisfied with the authors' solution. I think the authors have also clarified the novelty of their work. Well done.

Reviewer #3:

Remarks to the Author:

The authors addressed some of my previous concerns (e.g., a more complete theoretical perspective is provided, but it is unexplained and superficial). However, some of my concerns were not addressed (e.g., large vertebrate herbivores are not the only herbivores in grasslands), and others were superficially addressed (e.g., I do not trust expert knowledge guess-timates of herbivore abundances or their passing discussion of plant species effects). The failure to adequately address these concerns seriously weakens their claims.

In my opinion, as I previously stated, the bottom-line on this manuscript is that the authors attempt too much for the space limits of the journal. As I stated previously, they need to distill the paper into a simpler story; otherwise, the manuscript represents either a monograph length paper or several associated papers. This has to be one of the densest papers I have ever reviewed and is confusing without the reader devoting considerable effort to fathom the points. The theoretical foundations are not explained, and the expected patterns for the data are not explained except in one figure. The terms in the paper are not explanatory (e.g., fenced vs unfenced, why not with and without large herbivores).

The value of this study is lost in the constricted format of this journal, which does not serve the authors' work, the reader, or journal.

REVIEWER COMMENTS

Reviewer #1 (Remarks to the Author):

I commend the authors for their thorough work in addressing all reviewer comments. The trickiest issue was addressing estimates of grazing intensity. I am satisfied with the authors' solution. I think the authors have also clarified the novelty of their work. Well done.

##Thank you very much for your suggestions that have improved this paper – and for your positive assessment of the final product.

Reviewer #3 (Remarks to the Author):

The authors addressed some of my previous concerns (e.g., a more complete theoretical perspective is provided, but it is unexplained and superficial).

##We are glad you felt we addressed some of your previous concerns, including providing a more complete theoretical perspective.

##Re: your concern about our treatment of the theoretical literature – There have been a handful of reviews of the theory and models underlying both resource-control and consumer-control of plant biomass. Chase et al 2000 (*Ecology* 81, 2485-2497) and Gruner et al 2008 (*Ecology Letters* 11, 740-755), that we cite in the manuscript, not only review and compare these different theoretical lineages, but also provide example models. Importantly, resource-control models have been formulated in many different ways to represent various biological relationships that reduce consumption efficiency (see review in Gruner et al 2008). To explain the math behind these models, beyond the type of review already published in e.g. Gruner et al and Chase et al, would lead to a review of the differences among the formulations. However, these differences are not something we are able to test effectively with this experiment.

##We focused our text on the key difference in the predictions that emerges when comparing these bodies of consumer-control and resource-control theory: whether there is a super additive, additive, or sub additive impact of herbivores on the vegetation biomass. Our factorial experiment provides a strong test of this prediction. Additionally, our experimental dataset, with replication under many different site conditions, allows us to test contingencies associated with some of the mechanisms that have been invoked in this literature – the site conditions under which we see greater or weaker resource or consumer control (e.g., domestic livestock and gradients of soil N and precipitation) – which is an additional power of this work beyond the meta-analyses that have preceded it (including Chase et al and Gruner et al).

##In our manuscript and its revisions, we have focused on the core predictions about the additivity of the effects of resources and consumers that arise across the different model formulations. In particular, we have focused our introduction text on the key differences in the predictions of the body of resource- and consumer-controlled theory and added examples of biological drivers that could underpin the theory. The current introduction includes the text

In particular, “consumer-controlled” models predict that when consumers are limited by their food resources, consumption will increase to counter any additional production, leading to no net change in plant biomass²²”

AND “...“resource-controlled” theory that predicts increasing plant biomass along a productivity gradient even in the presence of herbivores²⁹⁻³¹. Importantly, this theory predicts that herbivores will consume a constant proportion of plant biomass, regardless of environmental productivity (Fig. 1d).”

In the previous revision, we also reframed, expanded, and rewrote the 3rd paragraph of the introduction to more explicitly link these bodies of theory to specific biological predictions about the strength of herbivore impacts and e.g., plant species turnover, soil nutrients, and climate – to provide examples of the many ways that grassland biomass could shift between consumer-controlled and resource-controlled. We particularly focused on biotic and abiotic factors that we can test to some extent using the empirical data from this experiment. We linked each of these examples with the conceptual/theoretical figure (Fig 1) that shows the associated *a priori* hypotheses that arise from these different bodies of theory (regardless of the model formulations).

##In the current version, we have further modified the paper to link to these predictions and tests. In the newly written Results section, we have now framed each section around these predictions (e.g., “Testing for an interaction between fertilization and fencing”; “Testing for an interaction contingent on herbivore type, herbivore biomass, or herbivory intensity”; “Testing for an interaction contingent on abiotic and biotic characteristics”). The Discussion, now separated from the Results, tackles and interprets these predictions in the context of the theoretical predictions about resource- and consumer-control. If you see additional ways that we can provide greater clarity and depth in our explanation and exploration of these predictions to address your point, please let me know. In particular, without a more specific understanding of this concern, it is difficult to know what additional explanation would be most helpful for you. If you have specific suggestions of what would provide greater clarity and depth beyond the published reviews we cite, the emergent predictions we describe, and the changes we have made to the text in this and the previous revision, we would be happy to try to address this.

However, some of my concerns were not addressed (e.g., large vertebrate herbivores are not the only herbivores in grasslands),

##We have added “vertebrate” to the abstract to clarify the focus of the experimental work. Previous work from this experiment (that we cited and discussed in the previous revision) shows that the fence effects we examine here are not biased by differential offtake by invertebrates. To further address this, we have added an additional sentence explaining the focus of the current work on vertebrates in the Methods. This section (including the sentence from the previous version followed by the newly added sentence) now reads: “Finally, although studies in some grasslands have shown arthropods can control plant biomass, we did not include insect herbivory in this study because previous work in this experiment suggests that arthropods increase in biomass with increasing plant biomass, but they do not strongly suppress plant biomass in any of the treatments⁵⁵. Thus, we focus here on the impacts of fences – and vertebrate herbivory – because evidence suggests that invertebrate herbivores are impacted by the treatments but have little overall impact on the treatments.”

and others were superficially addressed (e.g., I do not trust expert knowledge guess-timates of herbivore abundances or their passing discussion of plant species effects). The failure to adequately address these concerns seriously weakens their claims.

##We agree that it would be ideal to have data on measured herbivore biomass and consumption rates for each of the sites in this study. However, in the last revision version, we added and analyzed two different published methods for estimating site-level herbivore impacts that use entirely different data and assumptions to address reviewer concerns. In the current revision, have added text in the discussion that acknowledges limitations and also defines future data needs: “*While unavailable for the current study, site- and plot-scale measurements of the rates of plant productivity and consumption rates by vertebrate herbivores would provide additional insights into the global variation – and likely future trends – in herbivore control of grassland biomass.*” We also have added text to the Methods section acknowledging limitations of the metrics: “*While each of these provides information about potential and actual grazing intensity, neither is a direct site- or treatment-scale measure.*”

In my opinion, as I previously stated, the bottom-line on this manuscript is that the authors attempt too much for the space limits of the journal. As I stated previously, they need to distill the paper into a simpler story; otherwise, the manuscript represents either a monograph length paper or several associated papers. This has to be one of the densest papers I have ever reviewed and is confusing without the reader devoting considerable effort to fathom the points.

##Thank you for this summary. To address the comment about the text being too dense, we have now substantially revised the manuscript to separate the Results from the Discussion and added section subheadings into the Results section, as we described above, to simplify, streamline, and provide additional guideposts.

The theoretical foundations are not explained, and the expected patterns for the data are not explained except in one figure. The terms in the paper are not explanatory (e.g., fenced vs unfenced, why not with and without large herbivores). The value of this study is lost in the constricted format of this journal, which does not serve the authors’ work, the reader, or journal.

##We have responded to the concern about the theoretical basis and links to the predictions in our previous and current revisions and described them earlier in this response. One point we had not previously addressed is that we have chosen to use the terms “fenced” and “unfenced” when referring to the experimental treatments to be as clear and correct as possible in describing the actual manipulations.